# Use of Herbal Medications for Treatment of Osteoarthritis and Rheumatoid Arthritis

**DOI:** 10.3390/medicines7110067

**Published:** 2020-10-28

**Authors:** Breanna N. Lindler, Katelyn E. Long, Nancy A. Taylor, Wei Lei

**Affiliations:** Department of Pharmaceutical and Administrative Sciences, Presbyterian College School of Pharmacy, 307 North Broad Street, Clinton, SC 29325, USA; bnlindler@presby.edu (B.N.L.); kelong@presby.edu (K.E.L.); nataylor@presby.edu (N.A.T.)

**Keywords:** herb, osteoarthritis, rheumatoid arthritis, pain, inflammation

## Abstract

Arthritis is a chronic condition that affects nearly a quarter of the United States population. Osteoarthritis (OA) and rheumatoid arthritis (RA) are two major forms of arthritis associated with severe joint pain and reduced quality of life. Various pharmacological interventions may be utilized for arthritis treatment when non-pharmacological therapy is insufficient. However, pharmacological therapy can be associated with serious side effects and high costs. Therefore, alternative therapies have been under investigation. Herbal medications have shown the potential for safe and effective management of arthritis. For this review, we attempt to summarize the mechanisms, safety, and efficacy of herbal treatments for OA and RA. After searching electronic databases, we identified nine herbs among 23 clinical trials used for the treatment of OA or RA patients. Improvement of OA and RA symptoms, pain, and inflammation was demonstrated. The herbs exhibited strong anti-inflammatory and anti-oxidant activities, contributing to a reduction in inflammation and tissue damage. Several herbs elucidated new mechanisms for OA and RA treatment as well. Though these herbs have shown promise for OA and RA treatment, more studies and clinical trials are required for determining safety and efficacy, bioactivity, and optimal bioavailability.

## 1. Introduction

Arthritis is a common health issue that affects millions of people in the United States [1]. Patients suffering from arthritis struggle with severe joint pain and nearly half of all adults with arthritis experience persistent pain [1,2]. More than 100 types of arthritis have been identified [3]. Two of the most common types are osteoarthritis and rheumatoid arthritis. Both osteoarthritis and rheumatoid arthritis impair joint structure and function but differ in symptoms, pathophysiology, and treatment (Table 1).

Osteoarthritis (OA), also known as degenerative joint disease, is the most common form of arthritis [4]. OA is a biomechanical and inflammatory disease influenced by several factors such as mechanical and oxidative stress, injury, age, obesity, and metabolic disease [5]. OA is characterized by joint cartilage degeneration, changes in the underlying bone, and synovitis [6]. Pro-inflammatory and pro-catabolic mediators are found localized in synovial fluid and hydrolytic enzymes, such as matrix metalloproteinases (MMPs), are associated with cartilage degeneration. Extracellular matrix breakdown can trigger the accumulation of innate immune cells that lead to inflammation and tissue destruction [7]. Signaling pathways and responses, such as those involving nuclear factor κB (NF-κB) and mitogen-activated protein kinase (MAPK), have also been found to play a role [5]. OA has a slow onset, often beginning later in life and leading to disability. Symptoms include localized joint pain and tenderness as well as stiffness in the morning and after periods of activity.

Rheumatoid arthritis (RA) is a systemic condition involving immune dysregulation and inflammation, affecting multiple joints. Female gender, genetics, and smoking are risk factors for developing RA [8]. The presence of antibodies, or lack thereof, help classify RA into a seropositive or seronegative disease. Seronegative patients have more inflammation upon presentation, whereas seropositive patients have increased inflammation and joint damage over the course of the disease [8]. Extra-articular manifestations may be observed in seropositive cases or severe disease. Anti-citrullinated protein antibody (ACPA) perpetuates inflammation and is associated with bone erosions and pain [9]. The inflammatory nature of this disease eventually leads to permanent deformity. Overall, RA patients face a high rate of disability, with approximately 60% unable to work by at least 10 years after disease onset [10]. Symptoms of RA include tender, warm, and swollen joints as well as stiffness in the morning and from inactivity [11]. 

Despite modern disease-state knowledge, providing effective treatment for OA and RA is challenging. The American College of Rheumatology/Arthritis Foundation recommends current treatment options [12,13]. According to these guidelines, recommendations for OA include oral and topical non-steroidal anti-inflammatory drugs (NSAIDs), oral analgesics, serotonin, and norepinephrine reuptake inhibitors, and intra-articular corticosteroids [12]. The overall goal for RA treatment is to reduce pain and inflammation. Recommendations for RA are dependent upon early or established disease and disease activity levels. For early and established RA, traditional disease-modifying antirheumatic drug (DMARD) monotherapy, especially methotrexate (MTX), is strongly recommended for patients with low disease activity levels [13]. For RA patients already on DMARD monotherapy and with moderate or high disease activity, a traditional DMARD combination, a biologic, or tofacitinib is recommended. The mechanism of action of current drugs for OA and RA treatment has been reviewed previously [14].

Current pharmacotherapy provides options for alleviating pain and symptoms of OA and RA. However, the side effects associated with these treatments may limit their use. NSAIDs may be associated with gastrointestinal, cardiovascular, and nephrotoxic effects and have been excluded for long-term treatment of arthritis [15,16]. Acetaminophen can induce hepatotoxicity [17]. Tramadol can alter the gastrointestinal and central nervous system [18]. Intra-articular corticosteroids may have questionable efficacy for OA treatment and may further damage joints and tissues [19]. Hyaluronic acid injections provide OA pain relief with reasonable safety but may be expensive [19]. Non-biologic DMARDs are highly efficacious in early RA with low disease activity level; however, treatment with these drugs increase the risk of gastrointestinal disturbance, hepatotoxicity, nephrotoxicity, and blood disorders [20]. Biologics are effective for moderate to severe RA, but have low tolerability and increase the risk of serious infection, cancer, and heart failure [21]. Finally, Janus kinase inhibitors to manage moderate to severe RA, carry a risk of infection and blood disorders [22]. Ultimately, for both OA and RA, physicians and patients must weigh the benefits and risks of using pharmacological therapy. 

Concerns regarding the safety and costs of conventional arthritis therapies have sparked interest in natural remedies. In addition, difficulty with chronic pain management in arthritis has led to the investigation of herbal therapies. Herbs may offer a complementary or alternative method for effective and safe treatment. In this review, we summarize current pharmacological therapy utilized for OA and RA and we provide recent findings regarding herbal arthritis management. Specifically, we set out to describe the mechanisms, safety, and efficacy (including pain and inflammatory outcomes) of select herbal medications used for OA and RA.

## 2. Materials and Methods

Electronic databases including PubMed, Google Scholar, and ClinicalTrials.gov were searched using general keywords such as “herb’’, “arthritis”, “pain”, “inflammation” and “clinical trial”. We also conducted our search by using specific herb names. Only papers published after the year 2000 with full English text were included. We excluded clinical trials that tested formulations with 4 or more combined herbs. A total of 23 clinical trials including 9 herbs: *Boswellia* spp., *Curcuma* spp., *Eremostachys laciniata*, *Eucommia ulmoides*, *Matricaria chamomillia L.*, *Paeonia lactiflora*, *Tripterygium wilfordii Hook F*, *Withania somnifera*, and *Zingiber officinale* were reviewed. The results of these clinical trials are summarized in Table 2.

## 3. Results

A search for safer alternatives to pharmacological therapy in OA and RA management has gained momentum in the 21st century (Table 2). Natural medicines found all over the world are actively being investigated for their safety and efficacy. Recently, studies have reported novel mechanisms of herbal therapy in targeting arthritis pain and inflammation. Moreover, positive clinical outcomes have been reported in response to herbal treatment. This section will provide an overview of a significant role in treating OA and RA. 

### 3.1. Clinical Trials Using Single Herb

#### 3.1.1. *Boswellia* spp. 

Boswellia, also known as frankincense, has been used for centuries in traditional Ayurvedic medicine. This herb exhibits anti-inflammatory activity, including inhibition of microsomal prostaglandin E2 (PGE2) synthase-1 and 5-lipoxygenase, reducing production or activation of inflammatory mediators such as matrix metalloproteinase (MMP)-9, MMP-13, cyclooxygenase (COX)-2, and nitric oxide (NO), as well as analgesic and anti-arthritic effects [23,24]. Boswellia is thought to exert its beneficial effects on arthritis by improving the knee joint gap, reducing osteophytes, and attenuating inflammatory mediators, such as C-reactive protein and hyaluronic acid, associated with knee OA [25,26]. 

The safety and efficacy of *Boswellia serrata* have been investigated in several studies. Majeed et al. found that OA patients receiving oral *B. serrata* extract for 8 weeks demonstrated significant improvement in their Visual Analog Scale (VAS), Japanese Knee Osteoarthritis Measure (JKOM), and their Western Ontario and McMaster Universities Osteoarthritis Index (WOMAC) scores in comparison to placebo [26]. Another clinical trial found that oral supplementation of Boswellia extract for more than 4 months also significantly improved physical function by reducing pain and stiffness in OA patients compared to placebo, without causing serious adverse events [25]. Most recently, Razavi et al. found that the topical application of *Boswellia Carterii B.* (oliban oil) was associated with a significant improvement in OA pain and symptoms in comparison to control and diclofenac groups. However, activities of daily living, sport and recreation, and knee-related quality of life did not improve significantly [27]. 

#### 3.1.2. *Curcuma* spp. 

Roots of Curcuma are used as a spice commonly known as turmeric. Curcumin, a polyphenol extract of turmeric, is well known for its anti-inflammatory and antioxidant effects, and it has a long history of use in traditional Chinese and Ayurvedic medicine [28]. The anti-inflammatory activity of Curcuma may be attributed to multiple mechanisms [29]. Treatment with Curcuma has been found to strongly inhibit the production of inflammatory mediators, such as interleukin (IL)-1, tumor necrosis factor-alpha (TNF-α), IL-8, NO, and a variety of MMPs, via diminishing the activation of NF-κB, protein kinase B (Akt), and MAPK signaling pathways [29,30]. Curcuma has also demonstrated a COX-2 inhibition, leading to a reduction of prostaglandin synthesis [31]

Recently, Curcuma extracts and curcumin have been studied extensively for their anti-osteoarthritic effects. Kuptniratsaikul et al. demonstrated that patients randomized to *Curcuma domestica* experienced significant improvement in WOMAC scores in comparison to baseline over a 4-week period and less abdominal pain and discomfort compared to ibuprofen [32]. Another trial demonstrated that curcumin had comparable efficacy, but better safety and fewer adverse effects than diclofenac [33]. Curcumin was also associated with a weight-lowering effect, an anti-ulcer effect, and no requirement for histamine H2-receptor antagonists (H2 blockers) [33]. The short-term effects of highly-bioavailable curcumin (Theracurmin) on knee OA were investigated in a randomized, double-blind, placebo-controlled prospective study [34]. After 8 weeks, Theracurmin significantly reduced knee pain VAS scores and celecoxib dependence compared to placebo, with no significant effect on JKOM scores. Slight increases in triglycerides and creatinine as well as decreases in red blood cells and cholinesterase levels were observed in only a few patients [34]. A recent study reported that curcuminoids, including curcumin, demethoxycurcumin, and bisdemethoxycurcumin, combined with diclofenac showed a greater improvement in pain and functional capacity with better tolerability in patients with knee OA [31]. 

#### 3.1.3. *Eremostachys laciniata*

Decoctions of the roots and flowers of the Iranian herb, *Eremostachys laciniata*, is usually given to alleviate inflammatory conditions, including arthritis [35]. The mechanistic action of this herb is unclear. However, one study reported that the treatment with either crude methanol extract or fractions of *E. laciniate* resulted in a reduction in the inflammatory response induced by carrageenan in rat paw [36]. The aqueous extract of *E. laciniate* has also shown promising antioxidant activity as indicated by a strong DPPH radical-scavenging activity and reduction of H_2_O_2_- or HOCl-luminal chemiluminescence [37]. 

One study has investigated the effects of topical *E. laciniata* application on arthritis pain and symptoms [35]. This single-blinded, randomized clinical trial indicated that the application of 0.5% *E. laciniata* ointment to affected joints over 2 weeks significantly reduced VAS pain scores for arthritis and RA patients when compared to the application of control ointment. In comparison to a 0.5% piroxicam ointment, pain scores were lower for arthritis and RA patients applying *E. laciniata* for 1 week, indicating a better initial therapeutic response. However, after 2 weeks, pain scores were similar between *E. laciniata* and piroxicam groups. Over the whole study period, *E. laciniata* treatment was associated with a greater reduction in joint inflammation in comparison to piroxicam treatment. The researchers also determined that iridoid glycosides were the most prevalent isolates from their *E. laciniata* extracts and proposed that the anti-inflammatory activity observed in their arthritis patients may have been due to these compounds [35].

#### 3.1.4. *Eucommia ulmoides*

*Eucommia ulmoides* is an herb that has recently demonstrated potential for OA and RA treatment. Mechanistically, *E. ulmoides* has been found to reverse LPS-induced production of IL-1β, IL-6, TNF-α, inducible nitric oxide synthase (iNOS), and COX-2 via modulating the activation of toll-like receptor (TLR) 4 in murine macrophages [38,39]. *E. ulmoides* has also demonstrated a reduction in the production of IL-17, IL-1β, IL-6, MMP-3, and TNF-α by attenuating the activation of the phosphoinositol 3-kinase (PI3K)/Akt signaling pathway in OA and RA rat models [40,41]. Furthermore, an aqueous extract of *E. ulmoides* was found to reduce serum MMP-1, MMP-3, and MMP-13 and protect the articular cartilage in a rat OA model [42]. Lastly, aucubin, a bioactive component of *E. ulmoides*, has been found to reduce reactive oxygen species (ROS) production, caspase-3 activity, and cell apoptosis [43]. 

Most recently, Eucommia, in combination with meloxicam, exhibited greater effects on pain relief and patient satisfaction compared to the meloxicam monotherapy group [44]. Currently, a 12-week, multicenter, randomized, double-blind placebo-controlled clinical trial is underway to assess the safety and efficacy of an *E. ulmoides* extract in patients with mild OA [45].

**Table 2 medicines-07-00067-t002:** Recent clinical trials involving herbal medicines.

Herbs	Patient Groups	Dosage	Follow-up	Outcome of Intervention	Ref
***Boswellia* spp.**					
*Boswellia serrata* extract (3-acetyl-11-keto-β-boswellic acid and β-boswellic acid)	BSE *n* = 22; placebo *n* = 20	Boswellic acid 174.6 mg oral twice daily	120 days	Reduction of OA joint pain and stiffness; improvement in knee joint gap and osteophytes.	[25]
*Boswellia serrata* extract (Boswellin Super)	BS *n* = 24; placebo *n* = 24	Boswellin Super 100 mg oral daily	56 days	OA VAS, JKOM, and WOMAC improvements vs. placebo.	[26]
*Boswellia Carterri B* (Oliban oil)	oliban oil *n* = 51; diclofenac *n* = 51; sesame oil control *n* = 52	Oliban oil: 10 drops topically twice daily	42 days	Reduction in OA pain and symptoms vs. control and diclofenac. No effect on activities of daily living, sport and recreation, and knee-related quality of life.	[27]
***Curcuma* spp.**					
Curcuminoid complex	curmcuminoid complex + disclofenac *n* = 71; diclofenac *n* = 69	Diclofenac 50 mg with or without 500 mg curcuminoid complex, twice daily	28 days	Curcuminoid complex + diclofenac showed better improvement in KOOS subscales, viz. pain, quality of life, and less adverse effects vs. diclofenac group.	[31]
*Curcuma domestica* extract	CD extract *n* = 185; ibuprofen *n* = 182	CD extract 1500 mg/day; ibuprofen 1200 mg/day	28 days	Improvement of WOMAC pain and function scores and non-inferiority to ibuprofen for knee OA; significantly less abdominal pain and discomfort vs. ibuprofen.	[32]
Curcumin	curcumin *n* = 70; diclofenac *n* =69	Curcumin 1500 mg/day oral; Diclofenac 100 mg/day oral	28 days	Overall efficacy comparable to diclofenac for OA; weight-lowering, anti-ulcer effect, and reduced requirement for H2 blockers.	[33]
Curcumin (Theracurmin)	curcumin *n* = 18; placebo *n* = 23	Curcumin 180 mg/day oral	56 days	Reduced OA knee pain VAS scores, but no effect on JKOM scores vs. placebo; Reduced celecoxib dependence.	[34]
***Eremostachys laciniata***					
*Eremostachys laciniata*	EL *n* = 67; piroxicam *n* = 70	0.5% topical ointment	14 days	Reduction in VAS pain scores and joint inflammation for arthritis and RA vs. control and piroxicam; quicker therapeutic response than piroxicam at 1 week; comparable response to piroxicam at 2 weeks	[35]
***Eucommia ulmoides***					
*Eucommia*	Eucommia + meloxicam *n* = 70; meloxicam *n* = 70	Meloxicam 15 mg with or without 36 g Eucommia daily	28 days	Enhanced performance of meloxicam on the reduction of pain stiffness, and dysfunction.	[44]
*Cortex Eucommiae extract*	Recruiting	1 g/day of *E. ulmoides* Oliver extract	84 days	Ongoing.	[45]
***Matricaria chamomilla L.***					
*Matricaria chamomilla* oil	MC oil *n* = 28; diclofenac *n* = 28; placebo *n* = 28	30 mL of oil containing 600 g dried *chamomilla* flowers; 1% diclofenac gel	21 days	Reduction in need for acetaminophen.	[46]
*Chamomile* tea	chamomile tea *n* = 20; placebo *n* = 15	6 g/day Chamomile tea as two tea bags twice daily	42 days	Reduction in tender joints, ESR, and inflammation.	[47]
***Paeonia lactiflora***					
Total glucosides of paeony (TGP)	TGP *n* = 105; control *n* = 89	TGP group: TGP 1.8 g/day + MTX 10mg/week + LEF 20 mg/day; Control group: MTX 10 mg/week + LEF 20 mg/day	168 days	Reduced hepatotoxicity, alanine aminotransferase, and aspartate aminotransferase.	[6]
***Tripterygium wilfordii Hook F***					
*Tripterygium wilfordii* root ethyl acetate extracts	*n* = 13	dose escalations starting at 30mg/day up to highest tolerated dose or 570mg/day	112 days	EA extract of TwHF at doses > 360mg/day showed improvement in both clinical manifestations and laboratory findings.	[48]
*Tripterygium wilfordii* root ethanol/ethyl acetate extracts	placebo *n* = 12; low-dose *n* = 12; high-dose *n* = 11	placebo or low-dose (180 mg/day) or high-dose (360 mg/day) extract	140 days	Both low- and high-dose of TwHF extract showed improvement of symptoms and signs of inflammation and physical functioning and was well tolerated.	[49]
*Tripterygium wilfordii Hook F* extract	TwHF group *n* = 60; sulfasalazine *n* = 60	TwHF extract: 180 mg/day; sulfasalazine: 2 g/day	168 days	Higher rate for ACR 20, 50, and 70 response criteria; reduced IL-6 levels.	[50]
***Withania somnifera***					
*Withania somnifera* root aqueous extract	*n* = 11		8 days	Chondroprotective effect on damaged human osteoarthritis cartilage matrix.	[51]
*Withania somnifera* root aqueous extract	*W. somnifera* 250 mg *n* = 20; *W. somnifera* 125 mg *n* = 20; placebo *n* = 20	*W. somnifera* 250 mg or *W. somnifera* 125 mg or identical placebo twice daily	28 days	*W. somnifera* (both 125 and 250 mg) reduced pain, stiffness, disability, modified WOMAC score, and knee swelling index.	[52]
***Zingiber officinale***					
*Zingiber officinale* (Ginger) extract	ginger *n* = 103; placebo *n* = 101	zintoma capsules containing 250 mg powdered ginger	42 days	Reduced pain according to WOMAC; reduction in morning stiffness; no difference in side effects vs. placebo.	[53]
*Zingiber officinale* (Ginger) extracts in NLC	ginger *n* = 59; diclofenac *n* = 59	ginger extract in NLC with 5% ginger extract by weight; 1% diclofenac	84 days	Improved knee pain, stiffness, physical function, and PGA.	[54]
*Zingiber officinale* (Ginger) extract	ginger *n* = 37; rx drugs from specialist *n* = 40	1000 mg/day orally	84 days	Reduced pain scores; increased patient satisfaction.	[55]
*Zingiber officinale* (Ginger) extract	ginger extract *n* = 40; placebo *n* = 40; ibuprofen *n* = 40	ginger extract: 30 mg/day; ibuprofen: 3 × 400 mg/day	30 days	Reduced VAS scores and gelling or regressive pain.	[56]
Concentrated Ginger extract	ginger extract *n* = 124; placebo *n* = 123	2 × 255 mg of ginger extract; placebo: coconut oil	42 days	Reduced knee pain on standing and after walking, WOMAC index, and request of rescue medication; improved quality of life.	[57]

#### 3.1.5. *Matricaria chamomilla* L.

*Matricaria chamomilla*, also known as chamomile, has been used for centuries to treat joint pain [46,47]. The dried flower part of the plant has historically been used in the treatment of rheumatic pain and inflammation [47]. Now, chamomile is on the FDA’s “generally recognized as safe” herbs list [46]. As a member of the Asteraceae; Compositae family, Chamomile has two common varieties, German chamomile, and Roman chamomile [58]. The most popular formulation of chamomile is herbal tea [58]. Chamomile contains several phenolic compounds such as apigenin, quercetin, patuletin, luteolin, and glucosides. These compounds show anti-inflammatory action by reducing cytokines and PGE2, which play a role in the pathogenesis of arthritis [47,58]. 

A randomized, controlled clinical trial was performed to study the safety and efficacy of topical chamomile oil compared to diclofenac and placebo in patients with knee OA [46]. In comparison to diclofenac and placebo, chamomile significantly decreased the need for acetaminophen (the rescue drug) without adverse events, but it did not influence WOMAC questionnaire domain responses [46]. Another trial found that daily consumption of 6 g of chamomile tea was associated with a reduction in tender joints and erythrocyte sedimentation rate compared placebo for RA patients [47]. 

#### 3.1.6. *Paeonia lactiflora*


*Radix Paeonia*, the dried root of *P. lactiflora Pallas*, has a history of traditional use in Chinese medicine [59]. Decoctions of *Radix Paeoniae* have been used in the treatment of RA and other inflammatory/autoimmune disorders [59]. Water/ethanol extracts of *Radix Paeoniae Alba* contain total glucosides of paeony (TGP), consisting mainly of paeoniflorin [60]. Previous studies have demonstrated an inhibition of the production of PGE2, leukotriene B4, NO, ROS, and other pro-inflammatory mediators by TGP and paeoniflorin [59]. Paeonia has also demonstrated anti-inflammatory activity by reducing microvascular permeability and the infiltration of inflammatory cells [59]. The paeoniflorin component of TGP may also inhibit osteoclast differentiation and TNF-α-induced apoptosis via inhibition of NF-κB [61,62]. 

In a randomized clinical trial, the hepatotoxicity of a combination of methotrexate (MTX), leflunomide (LEF), and TGP with a combination of just MTX and LEF was compared [6]. At the midpoint and end of the study, the combination including TGP was shown to be just as efficacious, with significantly less hepatotoxicity. The hepatoprotective effects of TGP may be associated with its anti-inflammatory ability to reduce TNF-α, IL-6, and C-Reactive Protein (CRP) [60]. 

#### 3.1.7. *Tripterygium wilfordii Hook F*

*Tripterygium wilfordii Hook F* (TwHF) is a Chinese herb that has demonstrated immunosuppressive effects and has historically been used in the treatment of RA. Numerous preclinical studies have shown that extracts from the root of TwHF inhibit the expression of pro-inflammatory cytokines and mediators, adhesion molecules, and matrix metalloproteinases by macrophages, lymphocytes, synovial fibroblasts, and chondrocytes [63,64]. TwHF can also induce apoptosis in lymphocytes and synovial fibroblasts and inhibit their proliferation. The immunosuppressive, cartilage protective, and anti-inflammatory effects of TwHF extracts are well demonstrated, making it a good alternative for patients with RA refractory to conventional therapy [65]. 

Various extraction methods have been used in an effort to reduce TwHF toxicity and maximize its therapeutic benefit. In initial studies, extractions proved a therapeutic benefit, but frequent adverse events and occasional severe toxicities were reported [66,67,68]. However, one study found that an ethyl acetate (EA) extract (at doses between 180 mg/day and 570 mg/day) and a polyglycoside preparation inhibited joint swelling and suppressed adjuvant arthritis with less adverse events in comparison to other preparations [48,49]. In 2003, Cibere et al. reanalyzed the data from clinical trials conducted before 2002 and confirmed that topical *Tripterygium wilfordii* appeared to be efficacious in RA [69]. Lv et al. conducted a randomized controlled clinical trial to compare the efficacy and safety of TwHF with MTX in the treatment of RA patients for 24 weeks [70]. They found that patients treated with TwHF alone or in combination with MTX showed a better outcome in achieving the American College of Rheumatology (ACR) 20, 50, and 70 response criteria. A reduction in the 28-Joint Count Disease Activity Score was also shown. Furthermore, the benefits of TwHF monotherapy were observed up to 18 months after study completion [70,71]. Another study found that patients treated with TwHF extract demonstrated a higher response rate and a decreased expression of IL-6 compared to patients receiving sulfasalazine [50]. Several clinical trials have been completed just recently or are ongoing and examining the therapeutic benefits of TwHF in RA (ClinicalTrials.gov).

#### 3.1.8. *Withania somnifera*

*Withania somnifera* (Ashwagandha) is an Ayurvedic medicine known for its anti-inflammatory and analgesic effects. *W. somnifera* extract has been found to inhibit the production of TNF-α, IL-1β, and IL-12 by diminishing the activation of NF-κB and activator protein 1 (AP-1) signaling pathways [72]. *W. somnifera* extract also slowed the degradation of bovine Achilles tendon type I collagen by inhibiting the activity of collagenase [73]. Treatment with *W. somnifera* decreased swelling, redness, deformity, and ankylosis in a collagen-induced arthritis rat model [74]. The anti-arthritic activity of *W. somnifera* may be attributed to its ability to reduce ROS, TNF-α, IL-1B, IL-6, MMP-8, NF-κB activation, and increase IL-10 secretion [75]. 

Sumantran et al. demonstrated that an aqueous extract of *W. somnifera* showed a significant chondroprotective effect on damaged human OA cartilage via diminishing the gelatinase activity of collagenases [51]. A recent study has demonstrated its analgesic effects in patients with knee OA [52]. In this 12-week clinical trial, treatment with 125 or 250 mg of *W. somnifera* extract was associated with significant reductions in the mean WOMAC and Knee Swelling Index in comparison to baseline and placebo. A significant reduction in VAS scores for pain, stiffness, and disability was also observed. The higher dose showed efficacy earlier (at 4 weeks), better physician global assessments (excellent vs. good vs. fair), and less need for rescue medication with paracetamol compared to low dose and placebo [52]. 

#### 3.1.9. *Zingiber officinale*


*Zingiber offcinale*, commonly known as ginger, is strongly associated with relieving inflammatory symptoms [54]. The anti-inflammatory activities of ginger have been widely investigated in patients as well as in vitro and in vivo models. Treatment with ginger has reduced the production of PGE2, NO, IL-1β, IL-12, TNF-α, monocyte chemoattractant protein-1 (MCP-1), and regulated on activation, normal T cell expressed and secreted (RANTES) [76]. Ginger has also inhibited the expression of MHC class II molecules, interferon-gamma (IFN-γ), and IL-2 resulting in inhibition of the antigen-presenting activity of macrophages and T cell function [76]. Aryaeian et al. found that ginger inhibited T cell proliferation and activation by reducing the expression of T-bet and increasing GATA-3 production [77]. Further, studies demonstrated that ginger modulated the activation of NF-κB, COX-1, COX-2, peroxisome proliferator-activated receptor gamma (PPARγ), and lipoxygenase [53,54,77]. 

Several clinical trials have been performed to assess the effects of ginger on OA pain-relief. Haghighi et al. found that patients treated with 30 mg of ginger extract had lower VAS scores and gelling or regressive pain compared to placebo [56]. Similar results were reported by other clinical trials. One trial revealed that 250 mg ginger orally was superior to placebo at reducing VAS pain scores and morning stiffness [53]. Another study reported that two tablets of 500 mg of ginger daily improved VAS pain scores, significantly increased patient satisfaction, and was associated with no difference in side effects in comparison to control [55]. Another study revealed that ginger extract was associated with reducing knee pain and WOMAC indices, although it was associated with gastrointestinal side effects [57]. Later, Amorndoljai and colleagues found that topical ginger extract significantly improved knee pain, stiffness, physical function, and patient global assessments following 12 weeks of knee OA treatment [54,78]. 

### 3.2. Clinical Trials Involving Herbal Combinations 

There is evidence of synergistic effects when different herbs are combined, including improvement in therapeutic outcomes and safety. Oral supplementation with *B. serrata*, combined with N-acetyl-D-glucosamine and ginger for 6 months, was found to significantly improve pain-free walking distance and WOMAC signs/symptoms for patients with moderate knee OA in comparison to standard OA management, without safety or tolerability issues [79]. Boswellic acid combined with methylsulfonymethane has been found to improve knee OA pain management and functional recovery and reduce the intake of anti-inflammatory drugs [80]. In a 3-month randomized, placebo-controlled trial comparing the efficacy of curcumin and boswellic acid (Curamin) with curcumin (CuraMed) and placebo, Curamin significantly improved physical performance tests and WOMAC joint pain indices compared to placebo. Moreover, the effect size observed with the curcumin-boswellic complex was greater than that of curcumin alone [81]. Another study examined the effects of a combination of *C. longa* and *B. seratta* (CB). They found that CB (500 mg twice daily) was more effective than celecoxib (100 mg twice daily) as demonstrated by improvement in pain scores, walking distance, and joint line tenderness, with effects comparable to celecoxib for alleviating crepitus and increasing joint-range of movement [82]. In both curcumin-boswellia complex trials, tolerability was high, and no serious adverse events were reported [82,83]. In a multicenter, open-label study, a complex of *Curcumalonga*, *Harpagophytum procumbins* (Devil’s claw), and bromelain reduced VAS scores from baseline for acute and chronic OA patients and provided clinically relevant joint pain improvement, with greater tolerability and no serious adverse events [83]. The effects of ginger on arthritis have also been studied in combination with other herbs. A pilot study with a quasi-experimental design evaluated the effects of a ginger and *Acmella oleracea* complex on pain and inflammation after 4 weeks [84]. Data showed significant WOMAC, Tegner Lysholm Knee Scoring, physical activity, and fat-free mass improvements. There was a significant decrease in pain intensity in both knees as detected by VAS in both knees and no significant side effects [84].

### 3.3. Other Herbs with Potential for OA and RA Treatment

A few other medicinal plants have been investigated in preclinical studies using cellular and animal models, but have not been tested in clinical trials involving humans. Nevertheless, the mechanisms elucidated by these preclinical trials provide insight into the therapeutic potential of these natural medicines for OA and RA. Species of Angelica, Cinnamon, Glycyrrhiza, and Saposhnikovia have been widely used in traditional Chinese medicine and known for their anti-inflammatory and pain-relieving effects [85,86,87,88]. These herbs diminish the expression of inflammatory mediators, such as IL-1, IL-6, TNF-α, PGE2, and COX-2, via modulation of NF-κB activation and kinase activity in in vitro and in vivo arthritis models [88,89,90,91]. Furthermore, treatment with these herbs has been found to improve cartilage structure and texture, normalize bone remodeling, and inhibit osteoclastogenesis in animal arthritis models. Such findings suggest the beneficial use of these herbs for treating arthritis in human patients.

## 4. Discussion

In this review, we have summarized clinical trials demonstrating the effect of herbal medicines when used for OA and RA treatment. We found nine herbs that show promise in improving arthritis symptoms by providing pain relief and a reduction in the inflammatory response and oxidative stress (Table 3). The major mechanisms of action of these herbs include modulation of inflammatory signaling pathways and regulation of immune cell activity.

Though different in pathology and clinical symptoms, both OA and RA are associated with inflammation and oxidative stress. In the pathogenesis of the degenerative joint disease, OA, inflammatory cytokines, such as IL-1, IL-6, and TNF-α, promote the synthesis of collagenase and MMPs resulting in the degradation of collagen type II. RA is a systemic autoimmune disease, in which cytokines play critical roles in regulating the function of macrophages, T cells, and B cells. When used for arthritis, herbs have been found to provide anti-inflammatory and antioxidant activity. Such herbs inhibit pro-inflammatory signaling pathways, including NF-κB, MAPK, and Akt, and enhance anti-inflammatory pathways, such as PPARγ. Therefore, the synthesis of pro-inflammatory cytokines, chemokines, NO, and PGE2 are diminished, while the production of anti-inflammatory cytokines, such as IL-10 and TGFβ, are promoted. As a result, these herbs reduce the activation of immune cells and the activity/production of collagenase and MMPs. These changes can lead to an improvement in OA and RA symptoms. Herbs may also synergistically improve the efficacy of traditional drugs, such as MTX and diclofenac, with a low risk of adverse effects [31].

The herbal medicines that we reviewed may positively impact the management of OA and RA and may be associated with minimal adverse effects. However, several studies reported that herbs caused some side effects, such as hepatotoxicity, gastrointestinal upset (diarrhea, nausea, vomiting, and abdominal pain), dizziness, fatigue, cutaneous reactions, elevated levels of serum aminotransferase, male reproductive toxicity, and alterations in menstruation [31,41,92]. Future studies should investigate the appropriate dosage and duration of herbs for treating OA and RA. Low bioavailability, due to poor absorption, rapid metabolism, or elimination of active components, may also limit the use of herbal medicines [81]. Several approaches have been tested to increase the bioavailability of different herbs [78,81,93]. Recently, nanoparticle-based delivery systems have been applied to ginger extracts to increase its efficacy when used for OA treatment [78]. Mixed purified curcuminoids with turmeric volatile oil have also been found to increase the bioavailability of curcumin in human and animal models [81]. However, the bioavailability of other herbs utilized for OA and RA treatment is poor, and future studies are needed to develop techniques to overcome this issue.

## 5. Conclusions

Current pharmacological therapy options recommended for OA and RA are associated with variable efficacy and safety, especially for the treatment of chronic pain and inflammation. Certain herbal medicines may be used as a complementary therapy to work with or reduce the need for pharmacological agents. Treatment with herbal medicines may also offer a safer alternative with equal or superior efficacy. 

The anti-arthritic mechanisms of herbs include inhibition of pro-inflammatory and pro-catabolic mediators such as cytokines, PGE2, MMPs, ROS, apoptotic proteins via signaling pathways (NF-κB, RANKL, and PI3K/Akt). These activities may contribute to improvement in OA and RA joint pain, inflammation, swelling, structure, and function, with minimal adverse effects.

For future research, more trials are needed to determine the clinical safety and efficacy of herbal medicine in arthritis and other chronic pain conditions. Further studies on herbal chemical compounds and isolates may also help to provide more targeted therapy options. Lastly, the development of natural product formulations with ideal bioavailability and kinetics will be necessary for optimizing treatment.

## Figures and Tables

**Table 1 medicines-07-00067-t001:** Pathophysiology and current treatment of osteoarthritis and rheumatoid arthritis.

Disease	Major Risk Factors	Pathology	Clinical Manifestations	Available Therapies
Osteoarthritis	Age Gender Obesity/overweight Joint trauma/injury Genetics Bone deformities Metabolic disease (i.e., diabetes)	− Characterized by joint cartilage degeneration, changes in the underlying bone, and synovitis. − Pro-inflammatory mediators, such as IL-1β and TNFα, promote the immune responses, increase oxidative stress, inhibit the synthesis of type II collagen and proteoglycans, diminish the chondrocyte proliferation, and activate chondrocytes and synovial cells to produce MMPs. − MMPs induce the degradation of articular cartilage.	− Pain − Joint stiffness − Loss of flexibility − Tenderness − Grating sensation − Bone spurs − Swelling	− Non-steroidal anti-inflammatory drugs (NSAIDs) − Tramadol − Duloxetine − Corticosteroids − Hyaluronic acid
Rheumatoid arthritis	Genetics Smoking Gender Microbiome	− Characterized by synovial inflammation and hyperplasia, autoantibody production, cartilage and bone destruction, and systemic disorders. − Pro-inflammatory mediators attract the accumulation of immune cells and promote inflammation in the synovial membrane. − Chronic inflammation induces joint and tendon destruction leading to bone erosion.	− Pain, stiffness, tenderness, and swelling in more than one joint − Fever − Fatigue or tiredness − Weight loss − The same symptoms on both sides of the body	− Non-steroidal anti-inflammatory drugs (NSAIDs) − Disease-modifying antirheumatic drug: Methotrexate, leflunomide, hydroxychloroquine, and sulfasalazine − Biologics: TNF biologics and non-TNF biologics − Janus kinase (JAK) inhibitors: Tofacitinib and baricitinib

**Table 3 medicines-07-00067-t003:** Summary of herbal medications for osteoarthritis and rheumatoid arthritis treatment.

Disease	Herbal Medicine	Dosage (Per Day)	Treatment Time (Day)	Mechanism of Action	Clinical Implication (For both OA and RA treatment)
Osteoarthritis	*Boswellia* spp.	100, or 349.3 mg, or 10 drops (oil)	42–120	− Inhibit the production of inflammatory mediators, such as iNOS, COX-2, TNF-α, IL-1β, and ROS → reducing apoptosis of chondrocytes and synovial fibroblasts. − Reduce the production of MMP-1, MMP-3, MMP-9, and MMP-13 → slowing degradation of the extracellular matrix of cartilage and bone and reducing osteoclast formation and bone resorption. − Promote collagen synthesis.	− Herbal medications exhibit strong anti-inflammatory and antioxidative activities. − Herbal medications mimic anti-arthritis activities, as shown in the treatment of current medications, with fewer adverse effects. − Herbal medications could also enhance the anti-arthritis activities of current medications.
	*Curcuma* spp.	180 or 500 mg	28–56		
	*Eucommia ulmoides*	1 g extract or 36 g powder	28 or 84		
	*Matricaria chamomilla L.*	600 g dried flowers	21		
	*Withania somnifera*	125 or 250 mg	8 or 28		
	*Zingiber officinale*	Ginger powder: 250 or 1000 mg; Extract: 30, or 510 mg, or 5% of body weight	30–84		
Rheumatoid arthritis	*Eremostachys laciniata*	0.5% topical ointment	14	− Inhibit the production of PGE2, leukotriene B4, NO, ROS, and other pro-inflammatory mediators. − Induce apoptosis in lymphocytes and synovial fibroblasts and inhibit their proliferation. − Inhibit the production of histamine and bradykinin.	
	*Matricaria chamomilla L.*	6 g tea	42		
	*Paeonia*	1.8 g	168		
	*Tripterygium wilfordii Hook F*	30–570 mg	112–168		

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
