# Peer review of "Use of Herbal Medications for Treatment of Osteoarthritis and Rheumatoid Arthritis"

_medicines, 2020, doi:10.3390/medicines7110067_

Round 1

Reviewer 1 Report

Congratulations on your study.
After conducting an analysis and review of the manuscript, these are my suggestions, annotations and changes to make:

There has been a good introduction to the subject of the study and the data provided are good.

In the manuscript there is a good section on results but there is a poor research methodology. Since it is a review, a section called Material and Method should appear in which the methodology followed to obtain the results (all the clinical trials found) is explained. Also the procedure, the type of study to be searched, the databases, the inclusion criteria and exclusion criteria used, etc.

Remember to divide the study into sections: Abstract, Introduction, Material and Method, Results, Discussion, Conclusion, etc.

In the Abstract section, some more relevant data / result of the study should appear.
In Table 1. Replace the word Duration with the word Follow-up. In this same section (Tracking), put Days or Weeks or Months, do not put different data.

Good Conclusions section but there should be a Discussion section, in which the authors analyze the results obtained in their study.

I hope these changes are made.
Thank you very much

Author Response

Dear Reviewer,

On behalf of all co-authors, I would like to this opportunity to thank you for the constructive criticism and suggestions on our manuscript. The manuscript has been revised according to the reviewers’ comments and suggestions. Please find below our point-by-point responses to each of the raised concerns.

Note: the line numbers are indicated under the “All Markup” model in the “Track Changes”.

Congratulations on your study.
After conducting an analysis and review of the manuscript, these are my suggestions, annotations and changes to make:

There has been a good introduction to the subject of the study and the data provided are good.

Response:

We thank the Reviewer for the valuable comments and suggestions, which help us to improve the quality of the manuscript.

In the manuscript there is a good section on results but there is a poor research methodology. Since it is a review, a section called Material and Method should appear in which the methodology followed to obtain the results (all the clinical trials found) is explained. Also the procedure, the type of study to be searched, the databases, the inclusion criteria and exclusion criteria used, etc.

Response:

We thank the Reviewer for the suggestion. A “Material and Method” section (Line 90-97) has been added to discuss the methodology about collecting studies for this manuscript.

Remember to divide the study into sections: Abstract, Introduction, Material and Method, Results, Discussion, Conclusion, etc.

Response:

We thank the Reviewer for the constructive suggestion. We did re-organize information as suggested. In order to make the paper flows better, we merged the “Pathophysiology of Osteoarthritis and rheumatoid arthritis” and “Current Pharmacotherapy for OA and RA” into the Introduction (Add: Line 33-52 Line 56-77; Delete: Line 98-164). We also changed the title “Herbal Medicines for OA and RA: Clinical Trails” to “Results” (Line 165).

In the Abstract section, some more relevant data / result of the study should appear.

Response:

We thank the Reviewer for the suggestion. We added some more details about the data/results (Line 17-22).

In Table 1. Replace the word Duration with the word Follow-up. In this same section (Tracking), put Days or Weeks or Months, do not put different data.

Response:

We thank the Reviewer for the suggestions. We replaced “Duration” with “Follow-up” and used Days for all the studies included in the table (Line 246).  

Good Conclusions section but there should be a Discussion section, in which the authors analyze the results obtained in their study.

Response:

We thank the Reviewer for the suggestion. A “Discussion” section (Line 379-410) has been added to discuss the major findings, mechanisms, and limitations of herbal medicines.

I hope these changes are made.
Thank you very much

Reviewer 2 Report

The manuscript provides a nice overview regarding the potential use of different herbal medication for treating RA and OA patients.

However, the manuscript has few following limitations that need to be addressed

  • Authors discussed RA and OA simultaneously though the pathology, clinical symptom and involved mechanism driving these diseases are very different.
  • The manuscript appears bit superficial with very limited mechanistic aspects.
  • Authors did not adequately discuss how the different drugs thought to work.
  • Many of these herbal remedies simply might be working by reducing pain or combination of pain relief and protection of join damage by altering immune or bone or cartilage cells. Authors need to go bit deeper to outline what is known about those mechanistic aspects for the herbal medicines.
  • Authors need to provide a graphical representation of the unifying concepts for the paper for helping the readers grasp the concepts better.  
  • Authors also need to discuss what might be potential impact of over usage of these medication without comprehensive study for proper dosage.

Author Response

Dear Reviewer,

On behalf of all co-authors, I would like to this opportunity to thank you for the constructive criticism and suggestions on our manuscript. The manuscript has been revised according to the reviewers’ comments and suggestions. Please find below our point-by-point responses to each of the raised concerns.

Note: the line numbers are indicated under the “All Markup” model in the “Track Changes”.

The manuscript provides a nice overview regarding the potential use of different herbal medication for treating RA and OA patients.

Response:

We are very grateful to this Reviewer for his/her favorable comments and suggestions on our manuscript.

However, the manuscript has few following limitations that need to be addressed

  • Authors discussed RA and OA simultaneously though the pathology, clinical symptom and involved mechanism driving these diseases are very different.

Response:

We acknowledge the Reviewer’s suggestion very much. We discussed the OA and RA simultaneously since it has been done in numerous studies. It would be greatly appreciated if the reviewer could provide some specific suggestions on this issue, and we are willing to revise the manuscript to resolve this issue.

  • The manuscript appears bit superficial with very limited mechanistic aspects.

Response:

We thank the Reviewer for the constructive suggestion. We add some more details regarding the mechanisms of herbs on treatment of OA and RA. For each individual herb, we have one paragraph to describe mechanisms (Line 174-179, Line 192-200, Line 216-221, Line 233-241, Line 249-256, Line 264-272, Line 281-288, Line 305-312, and Line 322-332) and another paragraph to summarize the clinical trials in the revised manuscript.

  • Authors did not adequately discuss how the different drugs thought to work.

Response:

We thank the Reviewer’s comment. We are not sure whether the Reviewer concerns about the drugs for the current pharmacological therapy or the herbal medicines. If we are talking about the herbal medicines, we added more details about the mechanisms which provide some information how the herbs work on treatment of OA and RA (Line 174-179, Line 192-200, Line 216-221, Line 233-241, Line 249-256, Line 264-272, Line 281-288, Line 305-312, and Line 322-332). We also add a “Discussion” section in the paper and summarize mechanisms associated with the beneficial effect of herbs (Line 385-397). In the manuscript, we are trying to focus on the herbal medicines, and the mechanism of action of drugs for the current treatment of OA and RA has been reviewed in previous publications. Therefore, we do not discuss it in detail.

  • Many of these herbal remedies simply might be working by reducing pain or combination of pain relief and protection of join damage by altering immune or bone or cartilage cells. Authors need to go bit deeper to outline what is known about those mechanistic aspects for the herbal medicines.

Response:

We thank the Reviewer for the constructive suggestion. As mentioned above, we add some more details regarding the mechanisms of the herbal medicines.

  • Authors need to provide a graphical representation of the unifying concepts for the paper for helping the readers grasp the concepts better.  

Response:

We are grateful to Reviewer for the valuable suggestion. Instead of a graphical summary, we have summarized the major contents of our manuscript in a new “Discussion” section (Line 379-410), which would help the reader to understand the concepts.

  • Authors also need to discuss what might be potential impact of over usage of these medication without comprehensive study for proper dosage.

Response:

We thank the Reviewer for the suggestion. This is a great point. We appreciate the reviewer bringing this up. A paragraph (Line 398-410) in the “Discussion” section describe this issue.

Thank you so much.

Round 2

Reviewer 1 Report

The new sections of Materials and Methods and, discussion are good. This makes the study have a better methodology. The structural changes made to the text and the table make the manuscript clearer. Thank you for making the requested changes

Author Response

We thank the Reviewer for the constructive comments and suggestions. It is very helpful for improving the quality of this manuscript.

Reviewer 2 Report

The authors need to review the related literature regarding the pathology of RA and OA.

There are significant differences between these two diseases in terms of the involved molecular mechanisms, clinical manifestations, available therapies, and risk factors.

For example, in RA, bone erosion is one of the major pathological changes, whereas, for OA, osteophyte formation also occurs. There are many such major differences between these two diseases. Authors need to pay special attention to such differences and the similarities, clearly outlining those.

It is also important to note that there already multiple strategies available to treat RA patients. The authors also need to outline those and discuss how herbal medication can complement or replace some of those established treatment strategies for RA.

For OA, there are no effective approved treatment strategies available in terms of pharmacological drugs. Authors can emphasize this need and can rationalize how the complementary approach with herbal medicines can fill this void.

Author Response

We thank the Reviewer’s comments and suggestions. We add a Table 1 (Line 92) which summarizes the risk factors, pathology, clinical manifestations, available therapies. In the new Table 3 (Line 321), we summarize the clinical implications of herbal medications.